# Smoking cessation and counseling: A mixed methods study of pediatricians and parents

Tregony Simoneau[1]☯*, Jessica P. Hollenbach[2,3]☯, Christine R. Langton[3]☯, Chia-Ling Kuo[4], Michelle M. Cloutier[2]

1 Department of Pediatrics, Division of Pulmonary Medicine, Boston Children's Hospital, Boston, Massachusetts, United States of America, 2 Department of Pediatrics, UConn Health, Farmington, Connecticut, United States of America, 3 Asthma Center, Connecticut Children's Medical Center, Hartford, Connecticut, United States of America, 4 Department of Community Medicine and Health Care, Connecticut Convergence Institute for Translation in Regenerative Engineering, UConn Health, Farmington, Connecticut, United States of America

☯ These authors contributed equally to this work.
* Tregony.simoneau@childrens.harvard.edu

## Abstract

### Objective

Pediatric providers play an important role in parental and youth smoking cessation. The goal of this study was to understand smoking cessation attitudes of parents and the behaviors, confidence and self-efficacy of pediatricians related to providing smoking cessation counseling to parents and youth.

### Methods

A mixed methods study was conducted in a convenience sample of families (n = 1,549) and pediatric primary care clinicians (n = 95) in Connecticut using surveys and focus groups from April, 2016 to January, 2017.

### Results

The smoking rate (cigarettes or electronic cigarettes) among all households surveyed was 21%. Interest in quitting smoking was high (71%) and did not differ based on smoking amount, duration, type of community of residence (urban, rural, etc), or race/ethnicity. For example, compared to participants who smoked for <10 years, those who smoked ≥20 years had a similar interest in quitting (OR = 1.12; 95% CI: 0.85–1.48). Ninety percent of clinicians surveyed asked parents about their smoking behavior at least annually but 36% offered no smoking cessation counseling services or referral. Clinicians almost always reported counseling youth about the dangers of nicotine and tobacco use (99%), were more confident about counseling youth than parents (p<0.01) and reported low self-efficacy about smoking cessation and prevention counseling of parents and youth. Ninety-three percent of clinicians opined that electronic cigarettes were equally or more dangerous than cigarettes but 34% never counseled youth about the dangers of electronic cigarettes.

**Data Availability Statement:** All relevant data are within the manuscript and its Supporting Information files.

**Funding:** This work was supported by the National Heart, Lung, and Blood Institute (1U34 HL130665-01 (MMC)). The funders had no role in study design, data collection and analysis, decision to publish, or preparation of the manuscript.

**Competing interests:** The authors have declared that no competing interests exist.

## Conclusions

Clinicians frequently screen parents about their smoking behaviors, but rarely provide smoking cessation counseling and express low confidence in this activity. Clinicians are more confident counseling youth than parents. Clinicians also recognize the dangers of electronic cigarettes, yet they infrequently counsel youth about these dangers.

## Introduction

Pediatric health care providers can play an important role in parental and youth smoking cessation [1] by advising patients to avoid tobacco smoke exposure, by asking about parental smoking status and by referring patients and parents who smoke to smoking cessation programs [2]. Secondhand smoke exposure (SHSe) in children has been associated with an increased incidence of ear infections, lower respiratory tract infections, wheezing, asthma, and death from sudden infant death syndrome [3]. In children with asthma, SHSe is a common trigger of asthma symptoms, and has been shown to worsen asthma severity [4–6]. In addition to the impact of SHSe, parental smoking is also an important risk factor for adolescent smoking and 80% of adult tobacco users started smoking before the age of 18 years [7, 8]. Because of these significant health impacts, The American Academy of Pediatrics (AAP) recognized tobacco control as a strategic priority in 2005 and tobacco use as a pediatric disease [8]. However, as of 2017, there had been no significant increase in pediatrician-delivered advice related to smoking exposure or behavior since the release of this statement [9].

In addition to cigarette smoking, the use of electronic cigarettes (EC) has steadily increased over the past five years and is now the most commonly used tobacco product by adolescents [7]. EC may represent a new pathway to nicotine addiction for youth [10]. Therefore, assessing for SHSe, cigarette and EC use, and providing smoking cessation counseling and/or referral to youth and parents, are important aspects of preventive care in the pediatric visit.

The goal of this study was to understand the current tobacco and nicotine use behaviors of parents and the cigarette and EC prevention and cessation counseling practices of pediatricians in Connecticut to inform future efforts to develop effective tobacco and nicotine cessation/prevention counseling for parents and youth in the pediatric primary care setting.

## Materials and methods

### Study populations

**Parents.** An anonymous survey (S1 File) was self-administered to parents in the waiting room at 23 pediatric practices across Connecticut from April 2016 to January 2017. The survey was administered by the practice to any parent with a child being seen at the site with no exclusion criteria.

**Clinicians.** A 12-question Clinician Smoking Survey (S2 File) was completed by 95 clinicians at 32 pediatric practices throughout the state of CT, including 23 clinics where the Family Smoking Survey was also completed. In addition, three 60-minute focus groups were conducted with 34 clinicians from three pediatric urban clinics in Hartford, CT. Clinicians were provided with and reviewed a written informed consent document. The focus group note-taker documented the verbal consent provided by each participant, which was witnessed by the focus group facilitator. These same clinicians also completed the Clinician Smoking Survey prior to the focus group.

## Instruments

**Parent survey.** We developed our 12-question Family Smoking Survey in collaboration with the CT Chapter of the American Academy of Pediatrics (AAP) and the Department of Public Health Tobacco Control Program. We did not validate our survey but questions were adapted from Behavioral Risk Factor Surveillance System questions related to tobacco smoking. The first two questions asked whether anyone in the household currently smoked cigarettes or used electronic cigarettes (vaping). Those who answered "yes" to either of these questions continued with the survey and answered questions about attempts at quitting smoking/vaping, interest in quitting, and information desired about quitting, along with demographic information. Survey respondents were grouped by both home zip code location and practice zip code location, determined by The Five Connecticuts, which groups all of the towns and cities in CT based on population density, median family income and poverty, into five categories—wealthy, suburban, rural, urban periphery and urban core [11].

**Clinician survey.** Our Clinician Smoking Survey was modeled off of the American Academy of Pediatrics Periodic Survey #61 Tobacco Cessation Counseling [12] and was composed of questions related to pediatrician behaviors including 1) frequency of counseling parents who use tobacco about the importance of quitting; 2) providing referral to cessation programs (such as a quit line); 3) providing brief counseling to all youth to prevent tobacco initiation; and 4) screening all teenagers for tobacco and nicotine use and offer treatment [13]. In addition, clinicians were asked what services they offer to parents who smoke, age at which youth counseling is initiated, frequency of providing counseling directly to adolescents, self-efficacy in providing counseling and about reimbursement for tobacco cessation counseling activities.

**Clinician focus groups.** Each of the three focus groups was facilitated by an asthma specialist and explored the smoking and vaping counseling practices of the clinicians (S3 File). The focus groups used an open-ended question model, asking about the participants' knowledge of smoking cessation programs, ideas about what programs or materials would be useful to help parents with smoking cessation, educational strategies that would be helpful to prevent adolescents and youth from smoking, and finally, barriers to providing smoking cessation counseling to patients and families and what they needed as clinicians to overcome these barriers. The discussion was guided by the facilitator using the Delphi approach, attempting to reach group consensus for each of the areas [14].

The study was approved by the Connecticut Children's Institutional Review Board. Lunch was provided to the clinicians who participated in the focus groups and each participant received a $20 gift card as an incentive for participation.

**Statistical analysis of survey data.** Discrete and continuous variables were summarized by mean and standard deviation and categorical variables were summarized by frequencies and percentages. Goodness-of-fit tests were applied to test for representativeness of the sample based on residence by the Five Connecticuts and race/ethnicity. Each survey question was analyzed independently excluding subjects who didn't respond. Logistic regression was used to estimate odds ratios (OR) and 95% confidence intervals (CI) for the association between demographic and smoking/vaping characteristics with interest in quitting smoking or vaping. All statistical analyses were performed in R [15] and IBM SPSS Statistics, version 25, Armonk, NY.

**Analysis of clinician focus groups.** Focus groups were audio-recorded and transcribed verbatim. A note-taker was present to aid in the transcription and capture non-verbal cues. Readers were trained to extract themes from the focus groups using the template format described by Miles and Huberman [16]. Themes were coded independently by two study staff who prepared summaries of all the data on emergent themes. After independently coding the

transcripts, the two reviewers compared their findings, with any differences in coding adjudicated by a third reviewer (MMC).

# Results

## Parents

A total of 1,549 participants completed a survey (33% Hispanic, 53% Non-Hispanic White, 7% Non-Hispanic Black, and 7% Other, Table 1). As compared to the residents of CT, respondents

**Table 1. Demographics of survey participants, Connecticut, 2016–2017.**

| | Parents surveyed (n = 1,549) | Subset of parents who smoke or vape (n = 252) | Clinicians surveyed (n = 95) |
|---|---|---|---|
| **Characteristic** | n (%) | n (%) | n (%) |
| Households with reported cigarette | 261 (17%) | 204 (81%) | |
| Households with reported vape | 25 (2%) | 17 (7%) | |
| Households with cigarette and vape | 39 (3%) | 31 (12%) | |
| Households with neither cigarette or vape | 1224 (79%) | 0 (0%) | |
| Residence/Practice Location by Five Connecticut's | (n = 368)[a] | (n = 228)[a] | (n = 93)[a] |
| Wealthy | 0 (0%) | 0 (0%) | 5 (5%) |
| Suburban | 59 (16%) | 25 (11%) | 17 (18%) |
| Rural | 47 (13%) | 24 (11%) | 2 (2%) |
| Urban periphery | 132 (36%) | 95 (42%) | 31 (33%) |
| Urban core | 130 (35%) | 87 (38%) | 38 (41%) |
| Race/ethnicity | (n = 343)[a] | (n = 228)[a] | |
| Hispanic | 114 (33%) | 77 (34%) | |
| Non-Hispanic white | 180 (53%) | 124 (54%) | |
| Non-Hispanic black | 25 (7%) | 14 (6%) | |
| Non-Hispanic other | 24 (7%) | 13 (6%) | |
| Respondent age, years | (n = 351)[a] | (n = 221)[a] | |
| <25 | 83 (24%) | 39 (18%) | |
| 25–50 | 249 (71%) | 167 (76%) | |
| >50 | 19 (55) | 15 (7%) | |
| Location of smoke/vape | | (n = 222)[a] | |
| Car only | | 45 (20%) | |
| Home only | | 7 (3%) | |
| Car or home | | 18 (8%) | |
| Neither car or home | | 152 (68%) | |
| If interested in quitting, best way to get information about smoking cessation[b] | | (n = 152)[a] | |
| Quit line | | 30 (20%) | |
| Phone app | | 33 (22%) | |
| Physician's office | | 48 (32%) | |
| Brochure | | 28 (18%) | |
| Other | | 24 (16%) | |
| Clinician credentials | | | (n = 76)[a] |
| APRN/PA | | | 27 (36%) |
| MD/DO | | | 49 (64%) |

[a] Not all survey participants responded to question.

[b] Multiple responses possible.

were more likely to reside in the urban core (35% versus 19% in CT) and less likely to reside in wealthy and suburban towns (16% versus 32% in CT). In addition, Hispanic residents of CT were oversampled (33% versus 9% in CT).

Of the 1,549 participants, 300 (19%) reported at least one current cigarette smoker(s) and 64 (4%) had at least one current vaper(s) in the household. Two hundred fifty-two parents who completed the survey were themselves current smokers and/or vapers (16%). The majority of parents who smoked or vaped were between 25–50 years old, were Non-Hispanic White or Hispanic and lived primarily in urban core or urban periphery communities (Table 1). Of parents who smoked or vaped, 28% indicated that they smoked/vaped in the car and 11% smoked/vaped in the home (Table 1).

The majority of smokers (69%) reported smoking half a pack of cigarettes or less each day. Seventy-one percent (n = 152/213) of respondents who smoked and/or vaped were at least a little interested in quitting. Thirty-three percent (n = 83) had attempted to quit smoking or vaping within the last six months. Of smokers who had attempted to quit (n = 65), the most commonly used method of cessation was "cold turkey" (62%), while 25% used a smoking cessation aid and 14% used EC. Looking specifically at those who were both smokers and vapers, 56% reported EC as their method of cessation suggesting that these individuals were using ECs as their method of smoking cessation. Of those interested in quitting, parents most frequently reported that they would prefer to get information about how to quit from a physician (32%). The number of cigarettes smoked per day was similar between those individuals with no interest in quitting and those with some interest in quitting (Table 2). Likewise, interest in quitting did not vary by race/ethnicity or residence based on the Five Connecticuts (Table 2).

## Clinician survey

Ninety-five clinicians completed this survey. 90% of surveyed clinicians reported asking parents about smoking behaviors at least annually and 99% of clinicians reported counseling adolescents at least annually about the dangers of smoking cigarettes. Clinicians reported starting these counseling activities when youth were 11.6 ± 2.1 years of age. More than half (65%) of the clinicians counseled adolescents who smoked cigarettes about the dangers of EC at least annually, but only 57% counseled non-smoking adolescents about the dangers of EC at least annually. Thirty-six percent of the surveyed clinicians offered parents who smoked a referral for smoking cessation, 36% provided educational materials, but 36% offered no smoking cessation services to parents who smoked.

Reimbursement concerns did not contribute to the lack of smoking cessation counseling by the primary care clinicians as most clinicians (88%) said that reimbursement for counseling services played a small role or no role in influencing their smoking counseling activities. On the other hand, only 12 providers (13%) indicated that they knew how to code for tobacco-related counseling services.

Clinicians were more confident in counseling adolescents about smoking prevention and cessation than in counseling parents (Fig 1). However, only 2% of clinicians felt their counseling was more than somewhat effective.

## Clinician focus groups

In the focus groups, clinicians stated that smoking cessation was not a priority among the list of anticipatory guidance topics because of time constraints within a well-child visit. All clinicians endorsed that other issues such as gun violence, sex, and drugs were a higher priority given their time limitations (Table 3). Clinicians expressed a desire for an easy referral process and educational videos to help adolescents and parents quit smoking. Clinicians expressed

**Table 2. Association of demographic and smoking/vaping characteristics with interest in quitting smoking/vaping[a].**

| | Uninterested | Interested | Unadjusted Model |
|---|---|---|---|
| | n (%) | n (%) | OR (95% CI) |
| **Smokers[b]** | | | |
| Cigarettes smoked per day (n = 170) | | | |
| 1–9 | 17 (37.8) | 51 (40.8) | 1 [Reference] |
| 10–19 | 13 (28.9) | 43 (34.4) | 1.28 (0.60–2.74) |
| ≥ 20 | 15 (32.6) | 31 (24.8) | 0.80 (0.37–1.70) |
| Years smoked (n = 159) | | | |
| 1–9 | 15 (34.9) | 39 (33.6) | 1 [Reference] |
| 10–19 | 18 (41.9) | 42 (36.2) | 0.94 (0.46–1.91) |
| ≥ 20 | 10 (23.3) | 35 (30.2) | 1.12 (0.85–1.48) |
| Residence by 5 CT's (n = 193) | | | |
| Suburban | 5 (9.4) | 16 (11.4) | 1 [Reference] |
| Rural | 4 (7.5) | 14 (10.0) | 1.23 (0.30–5.00) |
| Urban periphery | 22 (41.5) | 52 (37.1) | 0.83 (0.31–2.24) |
| Urban core | 22 (41.5) | 58 (41.4) | 0.92 (0.34–2.49) |
| Race/ethnicity (n = 190) | | | |
| Non-Hispanic white | 30 (58.8) | 70 (50.4) | 1 [Reference] |
| Hispanic | 15 (29.4) | 53 (38.1) | 1.60 (0.79–3.23) |
| Non-Hispanic black or other | 6 (11.7) | 16 (11.5) | 1.21 (0.44–3.36) |
| **Vapers[c]** | | | |
| Ampoules/vials vaped per day (n = 17) | | | |
| <1 | 3 (50.0) | 0 (0.0) | 1 [Reference] |
| ≥ 1 | 3 (50.0) | 10 (100.0) | N/A |
| Number of years vaped (n = 27) | | | |
| 0–1 | 7 (77.8) | 16 (88.9) | 1 [Reference] |
| ≥ 2 | 2 (22.2) | 2 (11.1) | 0.44 (0.05–3.76) |
| Residence by 5 CT's (n = 40) | | | |
| Suburban | 2 (16.7) | 4 (14.3) | 1 [Reference] |
| Rural | 1 (8.3) | 8 (28.6) | 4.0 (0.27–58.56) |
| Urban periphery | 5 (41.7) | 13 (46.4) | 1.30 (0.18–9.47) |
| Urban core | 4 (33.3) | 3 (10.7) | 0.38 (0.04–3.61) |
| Race/ethnicity (n = 39) | | | |
| Non-Hispanic white | 7 (58.3) | 18 (66.7) | 1 [Reference] |
| Hispanic | 3 (25.0) | 5 (18.5) | 0.61 (0.12–3.27) |
| Non-Hispanic black or other | 2 (16.6) | 4 (14.8) | 0.74 (0.11–4.96) |

Abbreviations: OR, Odds Ratio; CI, Confidence Interval; N/A; Not Applicable

[a] Table includes 213 unique parents who responded to interest in quitting question, but Smoking and Vaping sections include 26 parents who both smoked and vaped but not all survey participants responded to question.

[b] Includes parents who only smoked(n = 173) or who smoked and vaped (n = 26).

[c] Includes parents who only vaped (n = 14) or who vaped and smoked (n = 26).

willingness to refer parents and youth to smoking prevention/cessation programs if such programs were available either in their practice or in the community. The clinicians referred parents to the CT Quit Line but were not aware of any smoking cessation programs available for adolescents. Clinicians indicated that the most teachable moment related to parental smoking cessation was in the newborn period and they would like training in how to be effective

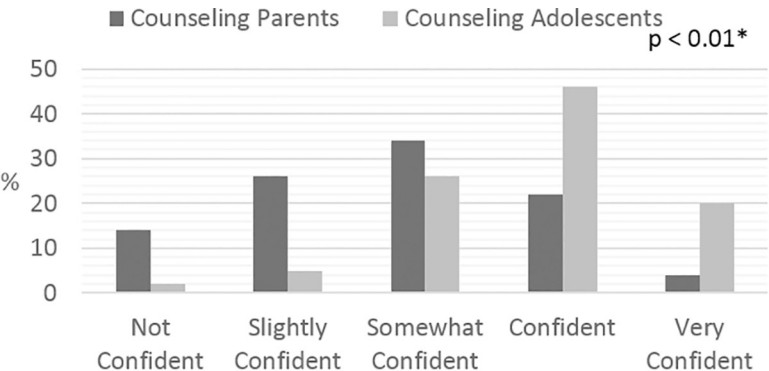

**Fig 1. Clinician confidence in counseling adolescents versus parents about smoking cessation.** Proportions of clinicians' confidence in counseling adolescent patients compared to counseling patients' parents regarding smoking cessation. * P value represents significant difference in clinician confidence between counseling parents versus adolescents (Fisher's exact test).

along with supporting materials. They also wanted a system by which obstetricians would inform the pediatricians of mothers who have stopped smoking during pregnancy and a program that they could implement for these mothers in the immediate postnatal period to prevent resumption of smoking. They also noted marijuana as a rising concern.

## E-cigarettes

The 48 respondents who vaped reported vaping for an average of 10.7 months and used 1.4 vials/day (range 0–4). On the clinician survey, almost all of the clinicians (92%) indicated that electronic cigarettes were equally or more dangerous and hazardous to health as compared to cigarettes, yet, 34% of providers never counseled their adolescent patients about the dangers of electronic cigarettes.

## Discussion

Use of nicotine-containing products remains a significant problem among households with children in CT. This study reports the attitudes and behaviors of parents and pediatric clinicians in CT related to tobacco use (both cigarette and electronic cigarette) and cessation counseling which will help to inform the design of effective programs and policies for facilitating smoking cessation in this population. This study confirms that the majority of parents who

**Table 3. Major themes identified from pediatric clinician focus groups.**

| Theme | Sample Comment |
|---|---|
| Clinicians had heard of the CT Quit Line, but not other smoking cessation programs. | *"I'll remember the quit line and bring in the quit line number for them"* |
| Smoking marijuana is more prevalent than smoking cigarettes. | *"The problem is not so much smoking cigarettes...The problem is smoking marijuana in my opinion"* |
| Clinicians identified a critical teaching window at the newborn visit where mothers who quit smoking while pregnant should be counseled to not resume smoking. They requested help from the OB/GYN providers with identifying who these mothers are. | *"...a lot of moms, really do quit all of those questionable habits while they're pregnant ... but the sustainability isn't there. If there was some way we could get a note that this [mom] has been actively participating in this [program] with their OB, we could carry it on."* |
| Clinicians requested quick, easy referral tools to assist with smoking cessation counseling as they have limited time in their visit to provide counseling themselves. | *"When it comes down to prioritizing the abuse, the food insecurity, the school failure, the eight hundred things, the parent that smokes usually winds up falling to the bottom"* |

smoke have at least some interest in quitting. However, almost one third of the parents who smoked had no interest in quitting. Identifying the subset of smokers who are not interested in quitting is important because it will require a different approach in terms of cessation counseling. Similarly, we confirmed that clinicians are beginning to counsel patients at an appropriate age (11.6 years) given that the mean start smoking age in CT youth is 13.9 years [17].

Similar to other studies, parents endorsed the important role of the doctor's office in obtaining information about quitting [18, 19]. While the clinicians in this study inquired about cigarette smoking yearly, they reported low self-efficacy about their current approach to smoking cessation and prevention with only 2% of clinicians reporting their counseling to be more than somewhat effective. This likely reflects a combination of factors including their limited time to complete a visit, along with a lack of easy referral methods. A major gap remains where parents would like information and support from their visits to the doctor's office, but the clinicians who need a quick and easy referral method do not provide information or support. This gap was noted by the AAP Periodic Survey comparing 2010 and 2004 pediatricians who were more likely to refer parents to a quit line, but less likely to plan a follow-up visit or call [9]. This problem continues today. Other studies have identified that both experience and exposure to formal training for smoking cessation increase clinician self-efficacy for smoking-cessation counseling [20]. Given the finding that clinicians reported greater confidence in counseling adolescents than parents, developing programming to increase clinician confidence and efficacy with counseling parents who smoke about how to create a smoke-free home and a smoke-free car is recommended.

The currently recommended approach to smoking cessation counseling is the 5As: ask, advise, assess, assist, and arrange. However, several studies have demonstrated that all five elements are rarely performed in the clinical setting [21]. The most difficult (and least frequently done) components are assist and arrange as they require knowledge about cessation tools and arrangement of follow-up so a simplified version—ask, advise, refer—has been developed [22]. The Health Plan Employer Data and Information Set (HEDIS) evaluates advise and assist as standard performance measures in their evaluation of managed care health plans as these are key components of smoking cessation counseling. However, this study found that pediatric clinicians frequently ask and advise, but rarely assist or arrange as indicated by their low rates of providing educational materials and referral to cessation programs [21]. In order to improve clinician delivery of smoking cessation counseling to parents and adolescents, systems that make the assist and arrange, or refer steps easier to perform are needed. In particular, education about specific resources, such as newer apps and websites (for example, quitSTART [23] or BecomeanEX.org [24]), could help clinicians provide effective assistance. There is also an opportunity to integrate the referral process through the Electronic Health Record (EHR) and have the Quitline call the patient/parent, as opposed to relying on the patient to call the Quitline. Termed Ask-Advise-Connect, this approach significantly increased the rate of enrollment in cessation treatment programs [25].

Clinicians agreed that smoking cessation counseling was important but they acknowledged that other issues are of higher priority than smoking cessation/prevention counseling. Lack of reimbursement was not a barrier raised by the clinicians. While clinicians mostly did not know how to bill for smoking cessation counseling, they did not indicate that this would alter their counseling. The CT Medicaid program reimburses clinicians $7.03 for 3–10 minutes of smoking cessation counseling [26], but this billing code must be associated with a nicotine dependency primary diagnosis code, making it challenging to bill for this service in the pediatric setting when providing counseling to the parents. When counseling a patient, however, the provider must submit a nicotine dependence ICD-10 code (F17.2) and then modify it with the appropriate procedure code for cessation counseling (99406 for 3–10 minutes, or 99406 for

over 10 minutes). That said, the clinicians indicated in the focus groups that they did not have three minutes to dedicate to smoking cessation counseling. One potential solution would be to use an identified person within the office, such as a nurse or a community health worker, to provide smoking cessation counseling and referrals to resources. This same person could then provide the assist and arrange components of the smoking cessation counseling by providing referral to a tobacco cessation program and arranging either a follow-up visit or phone call. There is currently very little data in the literature about the role of community health workers in smoking cessation counseling and this is an area for future study.

As seen in other studies, pregnancy was identified by clinicians as a motivator for quitting and an opportunity for intervention [27]. However, previous studies have also found high rates of relapse both during and after pregnancy [27]. Therefore, any intervention applied during pregnancy needs to span the entire pregnancy and postpartum period to avoid relapse.

Finally, this study also explored EC use by parents and the views of the clinicians toward EC use. Clinicians viewed EC as equally or more dangerous than smoking cigarettes. While the attitudes of youth toward EC were not explored in this study, others have found that the perception among youth is that ECs are safer than cigarettes [28]. Despite viewing EC as dangerous as cigarettes, primary care clinicians rarely reported counseling youth about the dangers of EC. This study did not investigate the reasons for this, yet other studies cite barriers such as lack of systematic screening, competing priorities, and limited confidence in a clinician's ability to council on EC use [29]. That said, if clinicians view EC as equally as dangerous as cigarettes, this may be difficult to reconcile with the potential harm reduction role of EC when they are used as a smoking cessation tool. In the focus groups, clinicians also indicated that marijuana smoking was a larger issue than cigarette smoking and reported that they felt unequipped to address this problem. They additionally voiced a general lack of knowledge about how to counsel about the adverse effects of smoking marijuana. While this was not the focus of this study and the surveys did not ask questions about marijuana smoking, this is an important area for future study, especially with the legalization of marijuana in several states. The recently published clinical report from the American Academy of Pediatrics begins to address some of these concerns [30]. Furthermore, the American Academy of Pediatrics' Richmond Center has many resources available for clinicians, patients, and families related to vaping, cigarettes, and marijuana [31].

Based on the results of this study and our review of the literature, efforts to assist families and youth around smoking cessation and prevention in the pediatrician's office should include: 1) Providing clinicians with education and tools about the dangers of EC; 2) Exploring the role of other health professionals in the office setting (nurses, medical assistants or community health workers) to support the busy clinicians in providing counseling and follow-up regarding smoking cessation and prevention with appropriate reimbursement for their time; and 3) Developing programs to identify pregnant mothers who quit smoking during their pregnancy and support them to remain smoke-free during the newborn "teachable moment".

## Limitations

The major limitations of this study are inherent to its design and include recall and social desirability bias as well as the relatively small sample size. In addition, the Family Smoking Survey was completed within practices, therefore selectively surveying people seeking care for their children. However, this is the population that could potentially be targeted with office-based programming and policy. Furthermore, our Family Survey did not ask participants to specify the type of EC utilized, the amount of liquid solution contained within the EC or if the solution contained nicotine. When our survey was implemented in 2016, sleek, high-tech EC's

with rechargeable batteries were entering the market and almost all products available contained some level of nicotine [32]. Additionally, inherent to survey studies, some of the questions may have been interpreted differently than we had intended and we acknowledge that additional field testing and validation of the survey would have improved the strength of the study. For example, "smoking cessation counseling" and "smoking cessation aids" were not clearly defined on either survey, leaving room for different interpretations by the families and clinicians completing the surveys. Focus groups were facilitated by an asthma specialist, which may have also biased the responses of the participants.

## Conclusions

In conclusion, parents in Connecticut who smoke are interested in quitting. Pediatric clinicians ask parents and youth about smoking behaviors but are more confident counseling youth than parents. The major reason for not counseling parents and youth is insufficient time because of the ever-increasing list of higher priority anticipatory guidance topics; facilitators to counseling include a quick and easy referral process. While clinicians acknowledge the dangers of EC, they counsel adolescents less frequently about EC than they counsel about the dangers of cigarettes.

## Author's note

This article was previously published in Journal of Community Medicine & Health Education [33], and was withdrawn on 12/7/2020. According to the corresponding author, the authors had requested withdrawal of the article from Journal of Community Medicine & Health Education before they submitted this work to PLOS ONE and were unaware that it had been published.

## Supporting information

**S1 File. Family smoking survey.** This survey was distributed to all families of children and youth seeking care at their pediatrician's office.
(DOCX)

**S2 File. Clinician smoking survey.** This survey was distributed to all pediatricians participating in the Easy Breathing asthma management program across Connecticut.
(DOCX)

**S3 File. Primary care focus group guide.** This focus group guide was used to facilitate discussions with pediatricians regarding their approach to somking prevention and cessation among their patients and families.
(DOCX)

## Acknowledgments

We thank the members of the U34 Smoking Working Group: Miguel Badillo, Rocio Chang, Martinus Evans, Jillian Wood and Barbara Walsh and the steering committee for *The Asthma Neighborhood*: *Collaborative for Asthma Equity (CASE) in Children*. We also thank Maria Thomas, Anita Hoey, Hilary Norcia and Mary Buckley-Davis, for distributing the Family Smoking Survey and the Clinician Smoking Survey. Additionally, we thank Autherene Mitchell for her help with creating the database and entering the data. Finally, we are indebted to the families and clinicians who completed the surveys.

## Author Contributions

**Conceptualization:** Tregony Simoneau, Jessica P. Hollenbach, Michelle M. Cloutier.

**Data curation:** Tregony Simoneau, Jessica P. Hollenbach, Christine R. Langton, Michelle M. Cloutier.

**Formal analysis:** Tregony Simoneau, Jessica P. Hollenbach, Christine R. Langton, Chia-Ling Kuo.

**Funding acquisition:** Michelle M. Cloutier.

**Investigation:** Tregony Simoneau, Michelle M. Cloutier.

**Methodology:** Tregony Simoneau, Jessica P. Hollenbach, Christine R. Langton, Michelle M. Cloutier.

**Resources:** Michelle M. Cloutier.

**Supervision:** Michelle M. Cloutier.

**Writing – original draft:** Tregony Simoneau.

**Writing – review & editing:** Jessica P. Hollenbach, Christine R. Langton, Chia-Ling Kuo, Michelle M. Cloutier.

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
