## [Decision Letter · Decision Letter 0]

22 Jan 2020

PONE-D-19-25711

Smoking Cessation and Counseling: A Mixed Methods Study of Pediatricians and Parents

PLOS ONE

Dear Dr Simoneau,

Thank you for submitting your manuscript to PLOS ONE. After careful consideration, we feel that it has merit but does not fully meet PLOS ONE’s publication criteria as it currently stands. Therefore, we invite you to submit a revised version of the manuscript that addresses the points raised during the review process.

We would appreciate receiving your revised manuscript by Mar 07 2020 11:59PM. To enhance the reproducibility of your results, we recommend that if applicable you deposit your laboratory protocols in protocols.io, where a protocol can be assigned its own identifier (DOI) such that it can be cited independently in the future. For instructions see: http://journals.plos.org/plosone/s/submission-guidelines#loc-laboratory-protocols

We look forward to receiving your revised manuscript.

Kind regards,

Stanton A. Glantz

Academic Editor

PLOS ONE

Journal Requirements:

2. Please state whether you validated the questionnaire prior to testing on study participants. Please provide details regarding the validation group within the methods section.

3. Please include a copy of the interview guide used in the study, in both the original language and English, as Supporting Information, or include a citation if it has been published previously.

4. Please provide additional details regarding participant consent for focus group participants. In the ethics statement in the Methods and online submission information, please ensure that you have specified (1) whether consent was informed and (2) what type you obtained (for instance, written or verbal, and if verbal, how it was documented and witnessed). If the need for consent was waived by the ethics committee, please include this information.

Reviewers' comments:

Reviewer's Responses to Questions

**Comments to the Author**

1. Is the manuscript technically sound, and do the data support the conclusions?

Reviewer #1: Yes

Reviewer #2: Yes

2. Has the statistical analysis been performed appropriately and rigorously? 

Reviewer #1: Yes

Reviewer #2: Yes

3. Have the authors made all data underlying the findings in their manuscript fully available?

Reviewer #1: Yes

Reviewer #2: No

4. Is the manuscript presented in an intelligible fashion and written in standard English?

Reviewer #1: Yes

Reviewer #2: Yes

5. Review Comments to the Author

Reviewer #1: This paper is a mixed-methods study of nicotine and tobacco use among parents and clinicians in pediatric settings. The authors found that while clinicians are screening parents for tobacco use, they are less confident in providing counseling to this group compared to youth. This paper is addressing a large number of topics and as a result, at times can seem unfocused. A clearer outline of the topics should be laid out in the introduction, and the same order should be following when displaying the methods, results and discussion. Some suggestions on how to clarify are provided below.

Introduction, General: The way the introduction is currently set up, the common theme among the topics is not easily understood. It would be more helpful if the first paragraph of the introduction started by talking about the importance of screening for and counseling about nicotine and tobacco use in the pediatric setting (e.g. moving the 3rd paragraph up).

Methods, Page 7, Lines 149-154: A little more explanation of the data analysis techniques would be helpful for the qualitative data. What was the coding process? (were some determined a priori, etc.). Was there any training in qualitative methods provided to the coders? Some discussion on inter-coder reliability would be helpful - what would happen in case of discrepancies? Was any computer software (e.g. Nvivo, Atlas.ti) used? Consider including the

Results, Page 8, Lines 171-173: It is not clear whether the questions about past attempts to quit were asked of cigarette smokers, vapers, or both. The fact that it mentions 23% used EC to quit suggests that this may be limited to just cigarette smokers? But if not, how was the question asked to people who use both EC and cigarettes (and what percentage of smokers were vaping?)?

Results, Page 8, Line 173: Does smoking cessation aid include both medications and behavioral support?

Results, Page 8, Lines 176-177: The terminology of “ampoules vaped per day” is a little confusing. Is this referring to cartridges? What about people who use e-cigarettes that use tank systems/are refilled with e-liquid? Or people who use disposable e-cigarettes? There doesn’t appear to be an assessment of what type of e-cigarette people are using, or whether that e-cigarette contains nicotine and if so what concentration, so this is difficult to interpret.

Results, Page 9, Lines 181-188: Were clinicians asked about smoking cessation counseling of parents in general, or were they asked separately about their practices involving cigarettes and e-cigarettes? Screening for and the comfort level with counseling for the two products may be different.

Discussion, Page 12, Lines 242-244: This may be a good opportunity to discuss more brief approaches to smoking cessation counseling, such as Ask, Advise, Refer that could be implemented in pediatric settings.

Table 1: Was data collected on the age of children from parents?

Table 3: It may be helpful to include the qualitative question guide as a supplement.

Reviewer #2: Summary: Dr. Simoneau presents an interesting paper detailing a mixed methods study of families’ and pediatricians’ behaviors around smoking and vaping in outpatient pediatric clinics in Connecticut. The study used surveys and focus groups to elucidate attitudes and barriers around smoking cessation.

Overall impression: There has been previous research highlighting the all-too-frequent disparity between physician beliefs and their actions regarding smoking cessation advice (https://www.ncbi.nlm.nih.gov/pubmed/17452234m, https://pediatrics.aappublications.org/content/140/1_MeetingAbstract/129). However, this paper presents a well-done study with a large sample of families. It is one of the first papers I have seen addressing vaping in this context.

Ways this paper could be strengthened even further:

-Include at least a rough estimate as to what percentage of caregivers refused to participate in the survey.

-Include the interview guide for the focus groups in the supplemental material.

-In Table 3, one of the themes was physicians needing easy referral tools to assist with smoking session, so in the discussion section, consider adding something about leveraging the electronic medical record or other technology for being able to make referrals to the quitline easier.

-Consider mentioning the major theme of marijuana smoking in the paper itself, not just in the table 3

Other thoughts:

-I wonder why only 2% of clinicians felt that their counseling was more than somewhat effective?

-Thank you for including the information about CT Medicaid program reimbursement; how should pediatricians then bill when providing cessation counseling for patients?

6. PLOS authors have the option to publish the peer review history of their article (what does this mean?). If published, this will include your full peer review and any attached files.

Reviewer #1: No

Reviewer #2: No

---

## [Author Response · Author response to Decision Letter 0]

4 May 2020

Response: We have addressed the style requirements.

2. Please state whether you validated the questionnaire prior to testing on study participants. Please provide details regarding the validation group within the methods section.

 Response: We did not validate our surveys. We have modified the Methods section as follows:

Parent Survey (pg. 5, line 108-109)

We did not validate our survey but it was similar to the Social Climate Survey of Tobacco Control that includes validated questions for assessing parental smoking behaviors [11].

Clinician Survey (pg. 6, lines 120-121) 

Our Clinician Smoking Survey was derived from the American Academy of Pediatrics tobacco cessation counseling survey and was composed of questions related to pediatrician behaviors including: ……

3. Please include a copy of the interview guide used in the study, in both the original language and English, as Supporting Information, or include a citation if it has been published previously.

 Response: We have included a copy of the Primary Care Clinician Focus Group Guide as supporting information (S3 File).

4. Please provide additional details regarding participant consent for focus group participants. In the ethics statement in the Methods and online submission information, please ensure that you have specified (1) whether consent was informed and (2) what type you obtained (for instance, written or verbal, and if verbal, how it was documented and witnessed). If the need for consent was waived by the ethics committee, please include this information.

 Response: Clinicians participating in our focus groups provided informed verbal consent. We have updated our methods section as follows: 

 Page 4, line 98-100: Clinicians were provided with and reviewed a written informed consent document. The focus group note-taker documented the verbal consent provided by each participant, which was witnessed by the focus group facilitator.

Response: For data not presented in tabular or graphic form, but presented in the body of the manuscript we have removed instances of the phrase “data not shown”. 

Reviewers' comments:

Reviewer's Responses to Questions

Comments to the Author

1. Is the manuscript technically sound, and do the data support the conclusions?

Reviewer #1: Yes

Reviewer #2: Yes

2. Has the statistical analysis been performed appropriately and rigorously? 

Reviewer #1: Yes

Reviewer #2: Yes

3. Have the authors made all data underlying the findings in their manuscript fully available?

Reviewer #1: Yes

Reviewer #2: No

Response: We have removed the two places which referred to “Data not shown” as this data is present in the manuscript, just not in tabular form. 

Data Availability: All relevant data are within the paper and its Supporting Information files. 

4. Is the manuscript presented in an intelligible fashion and written in standard English?

Reviewer #1: Yes

Reviewer #2: Yes

5. Review Comments to the Author

Reviewer #1: This paper is a mixed-methods study of nicotine and tobacco use among parents and clinicians in pediatric settings. The authors found that while clinicians are screening parents for tobacco use, they are less confident in providing counseling to this group compared to youth. This paper is addressing a large number of topics and as a result, at times can seem unfocused. A clearer outline of the topics should be laid out in the introduction, and the same order should be following when displaying the methods, results and discussion. Some suggestions on how to clarify are provided below.

Introduction, General: The way the introduction is currently set up, the common theme among the topics is not easily understood. It would be more helpful if the first paragraph of the introduction started by talking about the importance of screening for and counseling about nicotine and tobacco use in the pediatric setting (e.g. moving the 3rd paragraph up). 

Response: Thank you for this thoughtful comment. We have rearranged the introduction to place the emphasis on the importance of the pediatrician in screening for nicotine/tobacco use. In addition, we have changed the first sentence of the abstract. 

Methods, Page 7, Lines 149-154: A little more explanation of the data analysis techniques would be helpful for the qualitative data. What was the coding process? (were some determined a priori, etc.). Was there any training in qualitative methods provided to the coders? Some discussion on inter-coder reliability would be helpful - what would happen in case of discrepancies? Was any computer software (e.g. Nvivo, Atlas.ti) used? Consider including the

Response: We have updated our Methods section to include additional detail regarding our qualitative methods.

Page 7, lines 155-160: Focus groups were audio-recorded and transcribed verbatim. A note-taker was present to aid in the transcription and capture non-verbal cues. Readers were trained to extract themes from the focus groups using the template format described by Miles and Huberman [17]. Themes were coded independently by two study staff who prepared summaries of all the data on emergent themes. After independently coding the transcripts, the two reviewers compared their findings, with any differences in coding adjudicated by a third reviewer (MMC).

Results, Page 8, Lines 171-173: It is not clear whether the questions about past attempts to quit were asked of cigarette smokers, vapers, or both. The fact that it mentions 23% used EC to quit suggests that this may be limited to just cigarette smokers? But if not, how was the question asked to people who use both EC and cigarettes (and what percentage of smokers were vaping?)?

Response: As specified in our methods section, questions about past attempts to quit were asked of both cigarette smokers and vapers. People who reported using both cigarettes and EC were asked the same question about methods used to quit. While this made some of the response options of that question irrelevant, this was done in an effort to keep the survey brief. As shown in Table 1, 12% of smokers reported use of both cigarettes and EC. To clarify the cessation methods, we have added a sentence to the results. 

Page 9, line 186-188: Looking specifically at those who were both smokers and vapers, 56% reported EC as their method of cessation suggesting that these individuals were using ECs as their method of smoking cessation. 

Results, Page 8, Line 173: Does smoking cessation aid include both medications and behavioral support? 

Response: Smoking cessation aid was meant to refer to non-behavioral support. However, in retrospect, this may not have been clear on the survey as the response option was “Used smoking cessation aides (not e-cigarettes or vaping)”. The next two options listed were behavioral support options. We have added this inherent risk of survey studies to the limitations.

Page 17, line 353-355: Additionally, inherent to survey studies, some of the questions may have been interpreted differently than we had interpreted and we acknowledge that additional field testing and validation of the survey would have improved the strength of the study.

Results, Page 8, Lines 176-177: The terminology of “ampoules vaped per day” is a little confusing. Is this referring to cartridges? What about people who use e-cigarettes that use tank systems/are refilled with e-liquid? Or people who use disposable e-cigarettes? There doesn’t appear to be an assessment of what type of e-cigarette people are using, or whether that e-cigarette contains nicotine and if so what concentration, so this is difficult to interpret.

Response: We recognize that electronic nicotine devices (ENDs) and their technology evolve rapidly. At the time these surveys were administered (2016), the market was dominated by mod systems which used ampoules or vials to replace e-liquid and we were trying to use the most generic term to encompass all users and all devices. We have added this to our limitations. 

Page 17, Line 349-353: Furthermore, our Family Survey did not ask participants to specify the type of EC utilized, the amount of liquid solution contained within the EC or if the solution contained nicotine. In 2016, when we implemented our survey, sleek, high-tech EC’s with rechargeable batteries were entering the market and almost all products available contained some level of nicotine [32].

Results, Page 9, Lines 181-188: Were clinicians asked about smoking cessation counseling of parents in general, or were they asked separately about their practices involving cigarettes and e-cigarettes? Screening for and the comfort level with counseling for the two products may be different.

Response: Clinicians were asked what smoking cessation tools they offered parents. The clinicians were not asked separately about cessation tools for e-cigarettes. At the time of the survey (and as seen in the data), e-cigarettes were being marketed and used as cessation tools for cigarette smokers, so we did not think pediatricians would be counseling parents about how to quit vaping. Our understanding of the dangers of vaping has since changed. 

Discussion, Page 12, Lines 242-244: This may be a good opportunity to discuss more brief approaches to smoking cessation counseling, such as Ask, Advise, Refer that could be implemented in pediatric settings.

Response: Thank you for this suggestion. This has been added to the discussion along with mention of ask, advise, connect and use of the EHR to facilitate the referral process. 

Page 14, line 281-283: The most difficult (and least frequently done) components are assist and arrange as they require knowledge about cessation tools and arrangement of follow-up so a simplified version—ask, advise, refer—has been developed…

Page 14, line 292-296: There is also an opportunity to integrate the referral process through the Electronic Health Record (EHR) and have the Quitline call the patient/parent, as opposed to relying on the patient to call the Quitline. Termed Ask-Advise-Connect, this approach significantly increased the rate of enrollment in cessation treatment programs.

Table 1: Was data collected on the age of children from parents?

Response: No, we only collected age information for respondents.

Table 3: It may be helpful to include the qualitative question guide as a supplement.

Response: The clinician focus group guide has been included in Supplementary material, under S3 File.

Reviewer #2: Summary: Dr. Simoneau presents an interesting paper detailing a mixed methods study of families’ and pediatricians’ behaviors around smoking and vaping in outpatient pediatric clinics in Connecticut. The study used surveys and focus groups to elucidate attitudes and barriers around smoking cessation.

Overall impression: There has been previous research highlighting the all-too-frequent disparity between physician beliefs and their actions regarding smoking cessation advice (https://www.ncbi.nlm.nih.gov/pubmed/17452234m, https://pediatrics.aappublications.org/content/140/1_MeetingAbstract/129). However, this paper presents a well-done study with a large sample of families. It is one of the first papers I have seen addressing vaping in this context.

Ways this paper could be strengthened even further:

-Include at least a rough estimate as to what percentage of caregivers refused to participate in the survey.

Response: Unfortunately, we did not capture the proportion of families who did not complete the survey. Practices were encouraged to distribute surveys to all patients. We recognize that this is a limitation to our study.

-Include the interview guide for the focus groups in the supplemental material.

Response: The clinician focus group guide has been included in Supplementary material, under S3 File.

-In Table 3, one of the themes was physicians needing easy referral tools to assist with smoking session, so in the discussion section, consider adding something about leveraging the electronic medical record or other technology for being able to make referrals to the quitline easier.

Response: We appreciate this suggestion and have added this to the discussion.

Discussion, Page 14, Line 292-296: There is also an opportunity to integrate the referral process through the Electronic Health Record (EHR) and have the Quitline call the patient/parent, as opposed to relying on the patient to call the Quitline. Termed Ask-Advise-Connect, this approach significantly increased the rate of enrollment in cessation treatment programs [26]. 

-Consider mentioning the major theme of marijuana smoking in the paper itself, not just in the table 3 

Response: We agree with the reviewer that this is an important finding that, while not the focus of this study, should be addressed. We have added this to our discussion. 

Page 16, Line 327-334: In the focus groups, the clinicians also indicated that marijuana smoking was a larger issue than cigarette smoking and reported that they felt unequipped to address this problem. They additionally voiced a general lack of knowledge about how to counsel about the adverse effects of smoking marijuana. While this was not the focus of this study and the surveys did not ask questions about marijuana smoking, this is an important area for future studies to explore, especially with the legalization of marijuana in several states. The recently published clinical report from the American Academy of Pediatrics begins to address some of these concerns

Other thoughts:

-I wonder why only 2% of clinicians felt that their counseling was more than somewhat effective?

Response: We agree that this is remarkable and did get some insight from the focus groups as to why this is. Ultimately, it likely reflects the challenges of behavior change and need for close follow up that the pediatrician’s office is not equipped to provide, combined with the lack of time to complete the visit, and lack of referral resources. We have added a comment in the discussion.

Page 13, Line 264-268: While the clinicians in this study inquired about cigarette smoking yearly, they reported low self-efficacy about their current approach to smoking cessation and prevention with only 2% of clinicians reporting their counseling to be more than somewhat effective. This likely reflects a combination of factors including their limited time to complete a visit, along with a lack of easy referral methods.

-Thank you for including the information about CT Medicaid program reimbursement; how should pediatricians then bill when providing cessation counseling for patients?

Response: This is a helpful and practical suggestion and has been added to the discussion. 

Page 15, Line 305-308: When counseling a patient, however, the provider must submit a nicotine dependence ICD-10 code (F17.2) and then modify it with the appropriate procedure code for cessation counseling (99406 for 3-10 minutes, or 99406 for over 10 minutes).

---

## [Decision Letter · Decision Letter 1]

21 Jul 2020

PONE-D-19-25711R1

Smoking Cessation and Counseling: A Mixed Methods Study of Pediatricians and Parents

PLOS ONE

Dear Dr. Simoneau,

Thank you for submitting your manuscript to PLOS ONE. After careful consideration, we feel that it has merit but does not fully meet PLOS ONE’s publication criteria as it currently stands. Therefore, we invite you to submit a revised version of the manuscript that addresses the points raised during the review process.

The substance of the paper is now fine.  Per the suggestion of one of the reviewers, please report  odds ratios and 95% confidence intervals, rather than just showing the p values which does not indicate the direction of association.

This is the only change you need to make.

We look forward to receiving your revised manuscript.

Kind regards,

Stanton A. Glantz

Academic Editor

PLOS ONE

Reviewers' comments:

Reviewer's Responses to Questions

**Comments to the Author**

1. If the authors have adequately addressed your comments raised in a previous round of review and you feel that this manuscript is now acceptable for publication, you may indicate that here to bypass the “Comments to the Author” section, enter your conflict of interest statement in the “Confidential to Editor” section, and submit your "Accept" recommendation.

Reviewer #2: All comments have been addressed

Reviewer #3: All comments have been addressed

2. Is the manuscript technically sound, and do the data support the conclusions?

Reviewer #2: Yes

Reviewer #3: Yes

3. Has the statistical analysis been performed appropriately and rigorously? 

Reviewer #2: Yes

Reviewer #3: No

4. Have the authors made all data underlying the findings in their manuscript fully available?

Reviewer #2: Yes

Reviewer #3: Yes

5. Is the manuscript presented in an intelligible fashion and written in standard English?

Reviewer #2: Yes

Reviewer #3: Yes

6. Review Comments to the Author

Reviewer #2: Excellent job at responding to the comments. The manuscript mentions clinicians wanting more resources to help parents and teens quit smoking. My one final, emphatic suggestion would be to mention by name the AAP's Richmond Center: https://www.aap.org/en-us/advocacy-and-policy/aap-health-initiatives/Richmond-Center/Pages/default.aspx. The Richmond Center has so many great practical resources for pediatricians, and yet far too few pediatricians are aware of this up-to-date, well-curated resource.

Reviewer #3: The authors have addressed all the comments from the reviewers.

Few comments for further consideration:

1. The authors should consider adding the calculation of odds ratio and 95% confidence intervals, rather than just showing the p values which does not indicate the direction of association

2. Authors should also include harm reduction role of vaping while discussing the findings

7. PLOS authors have the option to publish the peer review history of their article (what does this mean?). If published, this will include your full peer review and any attached files.

Reviewer #2: No

Reviewer #3: No

---

## [Author Response · Author response to Decision Letter 1]

4 Sep 2020

The manuscript mentions clinicians wanting more resources to help parents and teens quit smoking. My one final, emphatic suggestion would be to mention by name the AAP's Richmond Center: https://www.aap.org/en-us/advocacy-and-policy/aap-health-initiatives/Richmond-Center/Pages/default.aspx. The Richmond Center has so many great practical resources for pediatricians, and yet far too few pediatricians are aware of this up-to-date, well-curated resource.

Thank you for this recommendation. We have added this resource to the discussion, page 16, line 334. We agree, it has a plethora of practical resources!

Few comments for further consideration:

1. The authors should consider adding the calculation of odds ratio and 95% confidence intervals, rather than just showing the p values which does not indicate the direction of association. 

We attempted to calculate odds ratios and 95% confidence intervals. Unfortunately, due to our small numbers with the vaping data, this resulted in some empty cells (see alternative Table 2 below). Given this, we felt that our original table with p values was more appropriate for our data. 

Table 2. Association of Demographic and Smoking/Vaping Characteristics With Interest in Quitting Smoking/Vaping

 Uninterested Interested Unadjusted Model

 n (%) n (%) OR (95% CI)

Smokers 

Cigarettes smoked per day (n=170) 

1-9 17 (37.8) 51 (40.8) 1 [Reference]

10-19 13 (28.9) 43 (34.4) 1.28 (0.60-2.74)

≥ 20 15 (32.6) 31 (24.8) 0.80 (0.37-1.70)

Years smoked (n=159) 

1-9 15 (34.9) 39 (33.6) 1 [Reference]

10-19 18 (41.9) 42 (36.2) 0.94 (0.46-1.91)

≥ 20 10 (23.3) 35 (30.2) 1.12 (0.85-1.48)

Residence by 5 CT’s (n=193) 

Suburban 5 (9.4) 16 (11.4) 1 [Reference]

Rural 4 (7.5) 14 (10.0) 1.23 (0.30-5.00)

Urban periphery 22 (41.5) 52 (37.1) 0.83 (0.31-2.24)

Urban core 22 (41.5) 58 (41.4) 0.92 (0.34-2.49)

Race/ethnicity (n=190) 

Non-Hispanic white 30 (58.8) 70 (50.4) 1 [Reference]

Hispanic 15 (29.4) 53 (38.1) 1.60 (0.79-3.23)

Non-Hispanic black or other 6 (11.7) 16 (11.5) 1.21 (0.44-3.36)

Vapersc 

Ampoules/vials vaped per day (n=17) 

<1 3 (50.0) 0 (0.0) 1 [Reference]

≥ 1 3 (50.0) 10 (100.0) N/A

Number of years vaped (n=27) 

0-1 7 (77.8) 16 (88.9) 1 [Reference]

≥ 2 2 (22.2) 2 (11.1) 0.44 (0.05-3.76)

Residence by 5 CT’s (n=40) 

Suburban 2 (16.7) 4 (14.3) 1 [Reference]

Rural 1 (8.3) 8 (28.6) 4.0 (0.27-58.56)

Urban periphery 5 (41.7) 13 (46.4) 1.30 (0.18-9.47)

Urban core 4 (33.3) 3 (10.7) 0.38 (0.04-3.61)

Race/ethnicity (n=39) 

Non-Hispanic white 7 (58.3) 18 (66.7) 1 [Reference]

Hispanic 3 (25.0) 5 (18.5) 0.61 (0.12-3.27)

Non-Hispanic black or other 2 (16.6) 4 (14.8) 0.74 (0.11-4.96)

Abbreviations: OR, Odds Ratio; CI, Confidence Interval; N/A; Not Applicable

If you feel otherwise and would like us to use this table instead of the one in the manuscript, please let me know. It does not change the text. 

2. Authors should also include harm reduction role of vaping while discussing the findings

This is an important point and we have added mention of this to the discussion on page 16, line 327.

---

## [Editor Report · Decision Letter 2]

28 Sep 2020

PONE-D-19-25711R2

Smoking Cessation and Counseling: A Mixed Methods Study of Pediatricians and Parents

PLOS ONE

Dear Dr. Simoneau,

Thank you for submitting your manuscript to PLOS ONE. After careful consideration, we feel that it has merit but does not fully meet PLOS ONE’s publication criteria as it currently stands. Therefore, we invite you to submit a revised version of the manuscript that addresses the points raised during the review process.

Please also report odds ratios and 95% confidence intervals, rather than just showing the p values which does not indicate the direction of association

We look forward to receiving your revised manuscript.

Kind regards,

Stanton A. Glantz

Academic Editor

PLOS ONE

---

## [Author Response · Author response to Decision Letter 2]

30 Oct 2020

Response to Reviewers:

Please also report odds ratios and 95% confidence intervals, rather than just showing the p values which does not indicate the direction of association

We have now replaced Table 2 to show odds ratios and 95% confidence intervals and made the necessary changes in the abstract and results.

---

## [Editor Report · Decision Letter 3]

18 Jan 2021

Smoking Cessation and Counseling: A Mixed Methods Study of Pediatricians and Parents

PONE-D-19-25711R3

Dear Dr. Simoneau,

We’re pleased to inform you that your manuscript has been judged scientifically suitable for publication and will be formally accepted for publication once it meets all outstanding technical requirements.

Kind regards,

Stanton A. Glantz

Academic Editor

PLOS ONE
---

## [Editor Report · Acceptance letter]

29 Jan 2021

PONE-D-19-25711R3 

Smoking Cessation and Counseling: A Mixed Methods Study of Pediatricians and Parents 

Dear Dr. Simoneau:

I'm pleased to inform you that your manuscript has been deemed suitable for publication in PLOS ONE. Congratulations! Your manuscript is now with our production department. 

Kind regards, 

on behalf of

Professor Stanton A. Glantz 

Academic Editor

PLOS ONE